# Customizing Language Models with Instance-wise LoRA for Sequential Recommendation

**Xiaoyu Kong**[1]  **Jiancan Wu**[1][*]**An Zhang**[2]
**Leheng Sheng**[2]  **Hui Lin**[3]  **Xiang Wang**[1]  **Xiangnan He**[1][*]
[1]MoE Key Lab of BIPC, University of Science and Technology of China
[2]National University of Singapore
[3]Electronic Science Research Institute of China Electronics Technology Group Corporation
kongxy@mail.ustc.edu.cn, wujcan@gmail.com
xiangnanhe@gmail.com

## Abstract

Sequential recommendation systems predict the next interaction item based on users' past interactions, aligning recommendations with individual preferences. Leveraging the strengths of Large Language Models (LLMs) in knowledge comprehension and reasoning, recent approaches are eager to apply LLMs to sequential recommendation. A common paradigm is converting user behavior sequences into instruction data, and fine-tuning the LLM with parameter-efficient fine-tuning (PEFT) methods like Low-Rank Adaption (LoRA). However, the uniform application of LoRA across diverse user behaviors is insufficient to capture individual variability, resulting in negative transfer between disparate sequences. To address these challenges, we propose Instance-wise LoRA (iLoRA). We innovatively treat the sequential recommendation task as a form of multi-task learning, integrating LoRA with the Mixture of Experts (MoE) framework. This approach encourages different experts to capture various aspects of user behavior. Additionally, we introduce a sequence representation guided gate function that generates customized expert participation weights for each user sequence, which allows dynamic parameter adjustment for instance-wise recommendations. In sequential recommendation, iLoRA achieves an average relative improvement of 11.4% over basic LoRA in the hit ratio metric, with less than a 1% relative increase in trainable parameters. Extensive experiments on three benchmark datasets demonstrate the effectiveness of iLoRA, highlighting its superior performance compared to existing methods in mitigating negative transfer and improving recommendation accuracy. Our data and code are available at https://github.com/AkaliKong/iLoRA.

## 1  Introduction

Sequential recommendation [1] suggests a user's next item of interest by analyzing his/her past interactions, tailoring recommendations to individual preferences. As Large Language Models (LLMs) [2] exhibit impressive proficiency in global knowledge comprehension and reasoning, their potential for application in sequential recommendation is garnering increasing interest [3–14]. Recent efforts [9, 10] approach the sequential recommendation task under a language generation paradigm, wherein user behavior sequences are converted into input prompts by either purely textual prompting (ID numbers or descriptions) [3, 5, 15, 6] or hybrid prompting with additional behavioral tokens [9, 10, 16], achieving remarkable success. Upon scrutinizing prior studies on LLM-based sequential recommenders, we can summarize a common fine-tuning pipeline comprising three components:

---

[*]Corresponding author

38th Conference on Neural Information Processing Systems (NeurIPS 2024).

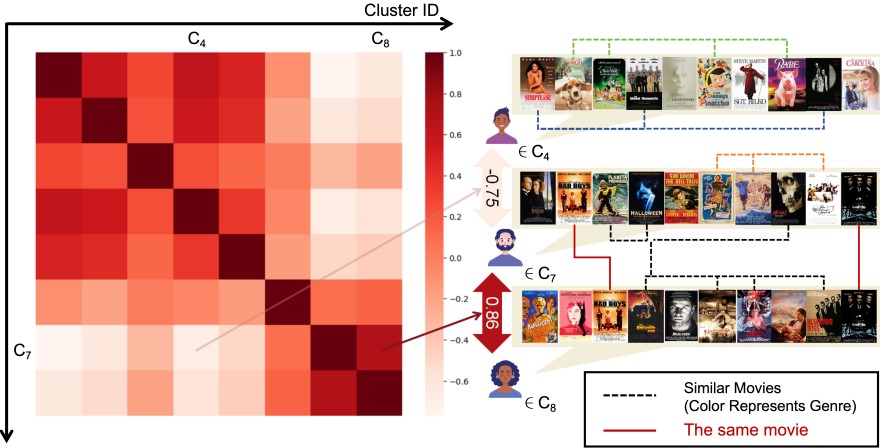

Figure 1: Gradient similarity of LoRA modules across training steps. The sequence dataset is partitioned into 8 clusters using Euclidean distance, with hierarchical clustering applied to reorder clusters, so that clusters closer in the collaborative space are also closer together in the heatmap. Gradient similarity is used to assess the geometric characteristics of the loss, with darker cells indicating higher similarity. In the case study on the right, dashed lines connect similar items, while solid lines link identical items. Users with a gradient similarity of 0.86 share a strong interest in thriller movies, while those with -0.75 cosine similarity show no clear preference alignment.

(1) convert a sequence of historical behaviors into a prompt; (2) pair these prompts with the subsequent items of interest, to create instruction-tuning datasets; (3) incorporate a trainable Low-Rank Adaptation (LoRA) module [5, 16, 10, 9] into LLMs and fine-tune it on such prompts. Existing studies [17, 3, 18, 6, 9, 10, 16] primarily focus on refining high-quality prompts to more effectively incorporate recommendation information, while leaving the choice of LoRA unexplored. Instead, they employ a standard LoRA module for fine-tuning, which freezes the LLM weights and updates the model through two additional low-rank matrices.

Previous work [19, 20] have demonstrated that related tasks tend to develop similar loss geometries, while unrelated tasks exhibit dissimilar ones. Using the same set of parameters in a multi-task learning context can lead to conflicts, especially when tasks have low gradient similarity. This can result in negative transfer, which limits further improvement of the model. From this perspective, since user behaviors often exhibit substantial individual variability (*e.g.,* distinct interests, behavior patterns, feedback mechanisms), we argue that employing a standard LoRA module across such diverse behaviors may not effectively capture these variabilities. Specifically, the inherent variability in user behaviors naturally inspires us to view item sequences with different behavior variables as different tasks. Simply applying a singular LoRA module leaves this multitask nature of sequential recommendation untouched, thus potentially overestimating the relationships between sequences. It easily causes the negative transfer between significantly discrepant sequences. Take LLaRA [9] as an example, which fine-tunes a LoRA module on top of Llama-2 [21] across all sequences. Following prior studies [19, 20], we use a symmetric heatmap to analyze the pair-wise gradients similarities of LLaRA associated with disparate sequences and reveal significant misalignment, as shown in Figure 1. Specifically, we observe strong clustering by membership closeness in the collaborative space, along the diagonal of the gradient similarity matrix. Meanwhile, clusters that are more distant in the collaborative space tend to exhibit dissimilar gradient trajectories. Clearly, such gradients are misaligned, which can result in suboptimal performance. To mitigate this issue, one straightforward solution is to deploy multiple LoRA modules, each fine-tuned for a specific sequence, enabling each module to act as a lightweight expert tailored to its respective sequence. However, it is impractical in terms of resources and time complexity, as the number of sequences often scales to millions.

To address these challenges, we propose a new fine-tuning framework, Instance-wise LoRA (iLoRA), which adapts the mixture of experts (MoE) concept [22, 23] to tailor the LLM for individual variability in sequential recommendation. The key idea is to integrate a diverse array of experts within the basic LoRA module, each encouraged to capture a specific aspect of user behaviors. Specifically, for each instance of item sequence a user adopted before, we feed it into a conventional recommender model (*e.g.,* SASRec [24]) to get a holistic sequence representation. Consequently, this representation,

reflecting personal behavior variability, is used by a gating network to customize instance-wise attention scores for the experts, where each score dictates the participation of each expert. With the attention scores, the experts are further assembled as instance-wise LoRA for this item sequence. Hereafter, we follow LLaRA [9] and fine-tune the LLM (*i.e.,* Llama-2 [21]) on a hybrid prompt, but instead with the personally-activated LoRA to mitigate the negative transfer between discrepant sequences. Importantly, iLoRA maintains the same total number of parameters as the standard LoRA, thereby avoiding overfitting while dynamically adapting to diverse user behaviors. We assess the effectiveness of iLoRA through extensive experiments on three benchmark sequential-recommendation datasets (*i.e.,* LastFM [25], MovieLens [26], Steam [27]), showcasing its superiority over leading methods (*e.g.,* GRU4Rec [28], Caser [29], SASRec [24], MoRec [30], TALLRec [5], LLaRA [9]).

## 2 Preliminary

**LLM-based Sequential Recommendation.** The primary task of sequential recommendation is to predict the next item that aligns with user preference [31, 32]. Formally, consider a user with a historical interaction sequence represented as $\mathbf{i}_{<n} = [i_1, i_2, ..., i_{n-1}]$, where each $i_j$ is an item interacted with at the $j$-th step. A sequential recommender, parameterized by $\boldsymbol{\theta}$, inputs this sequence and outputs a probability distribution over potential next items in the candidate set. This model is trained to maximize the likelihood of the true next item $i_n$:

$$\max_{\boldsymbol{\theta}} \sum_{\mathcal{D}} \log P_{\boldsymbol{\theta}}(i_n | \mathbf{i}_{<n}). \tag{1}$$

In the context of LLM-based sequential recommendation, we employ instruction tuning [33, 34], which fine-tunes LLMs using training data structured into explicit instructional pairs $(\mathbf{x}, \mathbf{y})$. Here, $\mathbf{x}$ comprises a detailed textual instruction describing the interaction sequences $\mathbf{i}_{<n}$ and recommendation task [17, 5, 15, 9, 16, 10], and $\mathbf{y}$ is the textual description of the predictive item $i_n$ in the user's sequence [35]. The training objective is formulated as an autoregressive model optimization problem:

$$\max_{\boldsymbol{\phi}} \sum_{(\mathbf{x}, \mathbf{y})} \sum_{t=1}^{|\mathbf{y}|} \log P_{\boldsymbol{\phi}}(y_t | \mathbf{x}, \mathbf{y}_{<t}), \tag{2}$$

where $\boldsymbol{\phi}$ denotes the LLM's model parameters, $y_t$ represents the $t$-th token in the output sequence, and $\mathbf{y}_{<t}$ includes all preceding tokens in the sequence. This objective ensures that each prediction is informed by both the prior items in the sequence and the detailed instructions describing the sequential recommendation task [32, 31, 9, 16, 10].

**Fine-tuning with Low-rank Adaption (LoRA).** Fully fine-tuning LLMs (*cf.* Equation (2)) entails substantial computational resources [36, 35, 37, 21, 38–40]. LoRA emerges as an efficient alternative [41–49], which injects trainable low-rank matrices into transformer layers to approximate the updates of pre-trained weights [46]. At the core, LoRA employs a low-rank decomposition where the update $\Delta \mathbf{W}$ to the pre-trained matrix $\mathbf{W} \in \mathbb{R}^{d_{\text{out}} \times d_{\text{in}}}$ is represented as $\Delta \mathbf{W} = \mathbf{B}\mathbf{A}$, where $\mathbf{B} \in \mathbb{R}^{d_{\text{out}} \times r}$ and $\mathbf{A} \in \mathbb{R}^{r \times d_{\text{in}}}$ and are tunrable up- and down-projection matrices, respectively. The rank $r$ is significantly smaller than both $d_{\text{in}}$ and $d_{\text{out}}$, enhancing adaptation efficiency.

Typically, LoRA applies such updates to the query and value projection matrices in the multi-head attention sub-layers within transformer layers [41]. Specifically, for an input $\mathbf{h}$ to the linear projection in the multi-head attention, LoRA results in the output $\mathbf{h}'$ as:

$$\mathbf{h}' = (\mathbf{W} + \frac{\alpha}{r} \Delta \mathbf{W})\mathbf{h} = \mathbf{W}\mathbf{h} + \frac{\alpha}{r} \mathbf{B}\mathbf{A}\mathbf{h}, \tag{3}$$

where $\mathbf{W}$ remains frozen, and $\alpha$ is introduced as a scaling factor that adjusts the influence of the updates *w.r.t.* the original $\mathbf{W}$.

This methodology introduces a flexible and efficient means to customize these models to new tasks, circumventing the need for extensive retraining of all model parameters.

**Fine-tuning with Hybrid Prompting.** In the field of LLM-based sequential recommendation, a critical challenge is the divergence between the natural language space and the "user behavior" space. To bridge this gap, previous research [9, 10, 16] introduces a hybrid prompt approach, which

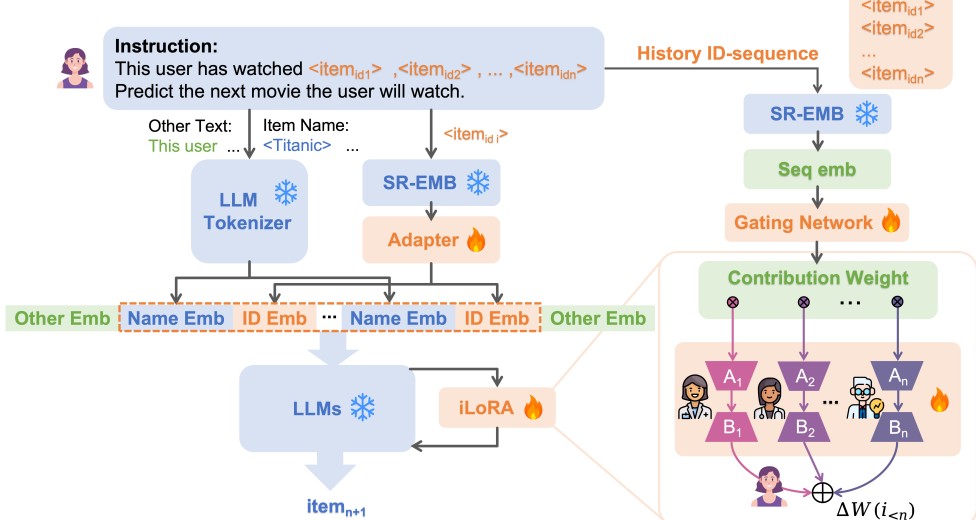

Figure 2: The iLoRA framework, which integrates the idea of MoE with LoRA, to implement sequence-customized activation patterns for various sequences.

incorporates behavioral insights captured by recommendation models into the prompts. This approach combines the textual token representation derived from the LLM's word embedding layer, with a behavior token representation learned from the recommender model with a cross-modal projector. Formally, for an item $i$ with associated metadata $txt$, the LLM tokenizer and word embedding layer LLM-TKZ$(\cdot)$ convert it into token representations $\mathbf{s}$:

$$\mathbf{s} = \text{LLM-TKZ}_{\phi}(txt). \tag{4}$$

Concurrently, item $i$'s ID embedding $\mathbf{z}$, pretrained using the encoder SR-EMB$(\cdot)$ of sequential recommender (*e.g.,* SASRec [24], GRU4Rec [28]), is represented as:

$$\mathbf{z} = \text{SR-EMB}_{\boldsymbol{\theta}}(i). \tag{5}$$

Then the ID embedding is transferred into a behavior token representation through a trainable projector Proj$(\cdot)$ parameterized by $\boldsymbol{\tau}_1$ to facilitate the alignment between the two modalities. This alignment strategy enables LLMs to interpret and leverage the behavioral knowledge distilled by conventional recommenders [9, 16, 10]. Subsequently, the textual and behavioral token representations are then concatenated to form a comprehensive description of item $i$:

$$\mathbf{e} = \text{Concat}(\mathbf{s}, \text{Proj}_{\boldsymbol{\tau}_1}(\mathbf{z})). \tag{6}$$

By converting each item into a hybrid token representation (*cf.* Equation (6)), we can rewrite the input prompt $\mathbf{x}$ describing the sequence $\mathbf{i}_{<n}$ and target response $\mathbf{y}$ decipting the item of interest $i_n$. Upon such prompts and responses, the objective of applying a uniform LoRA module is as follows:

$$\max_{\Delta\phi} \sum_{(\mathbf{x},\mathbf{y})} \sum_{t=1}^{|\mathbf{y}|} \log P_{\phi+\Delta\phi}(y_t|\mathbf{x}, \mathbf{y}_{<t}). \tag{7}$$

## 3 Methodology

To address the issue of negative transfer associated with conventional LoRA fine-tuning, we introduce the Instance-wise LoRA (iLoRA) fine-tuning framework. This innovative approach adapts the Mixture of Experts (MoE) concept [22, 23] to tailor Large Language Models (LLMs) to individual characteristics in sequential recommendation, as illustrated in Figure 2. At the core is the integration of multiple experts, each encouraged to capture a specific aspect of user behaviors. Different instances of user behavior (*i.e.,* item sequences) use a gating network to create instance-wise attention scores over experts. Such attentive experts instantiate the trainable matrices $\mathbf{B}$ and $\mathbf{A}$, thus personalizing a LoRA. Upon this instance with its individually activated LoRA, We fine-tune the LLM to minimize the negative transfer among disparate sequences.

## 3.1 Instance-wise Generation for Sequential Recommendation

Applying a uniform LoRA across the population of sequence instances risks overlooking individual variability and easily causes negative transfer, where distinct sequences might adversely affect each other. This inspires us to view the modeling of each individual instance as a separate task instead, and customize instance-wise LoRA module. By doing so, the LLM-based recommender is expected to align more closely with the behavioral and preference variability of individual users. Formally, for any sequence instance $\mathbf{i}_{<n}$, the autoregressive objective with instance-wise LoRA modules is as:

$$\max_{\Delta\phi} \sum_{(\mathbf{x},\mathbf{y})} \sum_{t=1}^{|\mathbf{y}|} \log P_{\phi+\Delta\phi(\mathbf{i}_{<n})}(y_t|\mathbf{x}, \mathbf{y}_{<t}), \tag{8}$$

where $\Delta\phi(\mathbf{i}_{<n})$ yields $\mathbf{i}_{<n}$-exclusive parameters of instance-wise LoRA, as compared to the shared parameters of uniform LoRA (*i.e.,* $\phi$ in Equation (7))

To this end, one straightforward solution is to set up different LoRA modules for individual sequences, enabling each module to act as the expert tailored to its respective sequence. However, it is impractical in terms of resource and time requirements, particularly as the number of sequences often reaches the millions. This highlights the need for a more scalable solution to address the challenge of sequence-specific customization without excessive computational overhead.

## 3.2 Instance-wise LoRA with the Mixture of Experts Concept

Instead of establishing various LoRA modules, we implement the mixture-of-experts (MoE) concept [22, 23] to devise our instance-wise LoRA (iLoRA) framework. This framework includes three components: (1) Diverging from the standard LoRA module with up- and down-projection matrices, we divide each matrix into an array of experts, each encouraged to capture a distinct, hidden aspect of user behavior; (2) For a given sequence instance, we use a gating network to obtain attention scores across the arrays of up- and down-projection experts, such that distinct sequences are likely to activate different experts; (3) Such an attentive combination of up- and down-projection experts instantiates the weights of LoRA, which are individually customized for the instance of interest. We will elaborate on these components one by one.

### 3.2.1 Splitting Low-Rank Matrices into Experts

Typically, the architectural foundation of LoRA is built upon two low-rank matrices: down-projection $\mathbf{B} \in \mathbb{R}^{d_{\text{out}} \times r}$ and up-projection $\mathbf{A} \in \mathbb{R}^{r \times d_{\text{in}}}$. Here we meticulously divide each projection matrix into an array of experts, as illustrated in Figure 2. Each expert is intended to focus on capturing one specific, hidden aspect of user preference. Formally, splitting the low-rank matrices is as follows:

$$\mathbf{B} = [\mathbf{B}_1, \mathbf{B}_2, \cdots, \mathbf{B}_K], \qquad \mathbf{A} = [\mathbf{A}_1, \mathbf{A}_2, \cdots, \mathbf{A}_K], \tag{9}$$

where $\mathbf{B}_k \in \mathbb{R}^{d_{\text{out}} \times r^*}$ and $\mathbf{A}_k \in \mathbb{R}^{r^* \times d_{\text{in}}}$ are the up- and down-projection pairs for the $k$-th expert, respectively; $r^* = \frac{r}{K}$ is the partial rank determined by the total rank $r$ of LoRA and a predefined number of experts $K$.

By dividing individual LoRA modules into specialized experts, we ensure a more granular and precise adaptation to user preferences. This segmentation approach allows each expert to focus on specific aspects of user interaction patterns, thereby mitigating the risk of negative transfer that arises from generalized adaptations. We should emphasize that such a segmentation scheme preserves the overall number of parameters equivalent to that of the standard LoRA, therefore preventing the potential overfitting issue.

### 3.2.2 Generating Instance-wise Attentions over Experts

Having obtained the experts (*i.e.,* up-projection submatrices $\{\mathbf{B}_k\}_{k=1}^{K}$, down-projection submatrices $\{\mathbf{A}_k\}_{k=1}^{K}$), we construct an instance-guided gating function to yield the contribution of each expert tailored to a specific sequence. Specifically, for a sequence of historical items $\mathbf{i}_{<n} = [i_1, i_2, \cdots, i_{n-1}]$, we utilize a sequential recommender (*e.g.,* SASRec [24]) to extract its representation as follows:

$$\mathbf{z} = \text{SR-EMB}_{\boldsymbol{\theta}}(\mathbf{i}_{<n}). \tag{10}$$

Here $\mathbf{z} \in \mathbb{R}^d$ provides a holistic view of the user's behavioral patterns and preferences. Subsequently, to ascertain the influence of each expert on distilling behavior patterns from this sequence, we get the contribution scores via a linear transformation with a softmax function:

$$\omega = \mathrm{Softmax}(\mathrm{Proj}_{\tau_2}(\mathbf{z})), \tag{11}$$

where $\mathrm{Proj}_{\tau_2}(\mathbf{z}) = \mathbf{W}_g \mathbf{z}$ with the trainable transformation matrix $\mathbf{W}_g \in \mathbb{R}^{K \times d}$, and the softmax function ensures these contributions normalized as the attention scores, thereby preventing any single expert from disproportionately influencing the recommendation; $\omega$ represents the attention scores over all the experts, with the $k$-th element indicating the contribution of expert $K$.

By using the sequence representation as the guidance signal, we can get the instance-wise attention scores over experts and encourage each expert's contribution closely aligned with the individual variability inherent in the sequence. Moreover, distinct sequences tend to yield different attention scores and activate different experts, while similar sequences incline toward analogous attention scores. By dynamically adapting to a wide range of user behaviors and preferences, these experts could specialize in diverse aspects of user behaviors and be more adept at handling diverse user needs.

### 3.2.3 Aggregating Mixture of Experts as Instance-wise LoRA

For the instance $\mathbf{i}_{<n}$ associated with the instance-wise attentions $\omega$, we use the mixture-of-experts concept to aggregate the up- and down-projection submatrices from different experts, so as to establish the instance-wise LoRA parameters:

$$\Delta \mathbf{W}(\mathbf{i}_{<n}) = \sum_{k=1}^{K} \omega_k \mathbf{B}_k \mathbf{A}_k, \tag{12}$$

where $\Delta \mathbf{W}(\mathbf{i}_{<n})$ encapsulates the adjustments enabled by our iLoRA. The attention score $\omega_k$, assigned by the gating network (*cf.* Equation (11)), reflects the relevance of expert $k$'s contribution to the particular sequence.

We apply such instance-wise LoRA updates on the transformer layers of the base LLM, which collectively construct the tunable parameters $\Delta \phi(\mathbf{i}_{<n})$. Clearly, iLoRA maintains the same total number of parameters as the standard LoRA, but dynamically customizes varying LoRA modules for different instances. This dynamic adaptation of parameters ensures that our model remains flexible and responsive to the varied preferences and behaviors exhibited by users, effectively managing the complexity inherent in sequential recommendation systems. This approach not only enhances personalization but also improves the predictive accuracy of the recommendation system.

## 4 Experiments

In this section, we first justify the need to reshape the fine-tuning task with a uniform LoRA module as a multi-task learning framework for sequential recommendation. We request the dataset from LLaRA [9] and maintain exactly the same experimental settings as described in the original paper. Our study builds upon the main table from the LLaRA paper, using the results reported in the LLaRA paper directly. Here we conduct extensive experiments on various real-world datasets, including LastFM [25], MovieLens [26], and Steam [27], to evaluate the effectiveness of our iLoRA framework. Our analysis includes detailed comparisons of iLoRA against established baseline models, which encompass both traditional sequential recommender models (*e.g.,* GRU4Rec [28], Caser [29], SASRec [24]) and LLM-based recommender models (*e.g.,* Llama2-7B [21], GPT-4 [50], MoRec [30], TALLRec [5], LLaRA [9]). ValidRatio [9] and HitRatio@1 are used as evaluation metrics, to separately quantify the ratios of valid responses over all sequences and relevant items over all candidate items, reflecting the model capability of instruction following and recommendation accuracy. See Appendix A for more details of these baselines, datasets, and metrics. Moreover, we perform a thorough ablation study to identify the key components that enhance iLoRA's performance, focusing particularly on the role of the gating network and expert settings. In a nutshell, we would like to answer the following research questions:

- **RQ1:** What is the rationale behind instance-wise LoRA compared to the uniform LoRA module?
- **RQ2:** How does iLoRA perform in comparison to traditional sequential recommender systems and LLM-based recommender models?

Table 1: The Results of iLoRA compared with traditional sequential recommender models and LLMs-based methods.

| | LastFM | | | MovieLens | | | Steam | | |
|---|---|---|---|---|---|---|---|---|---|
| | ValidRatio | HitRatio@1 | Imp.% | ValidRatio | HitRatio@1 | Imp.% | ValidRatio | HitRatio@1 | Imp.% |
| **Traditional** | | | | | | | | | |
| GRU4Rec | 1.0000 | 0.2616 | 91.13% | 1.0000 | 0.3750 | 40.67% | 1.0000 | 0.4168 | 26.30% |
| Caser | 1.0000 | 0.2233 | 123.91% | 1.0000 | 0.3861 | 36.62% | 1.0000 | 0.4368 | 20.51% |
| SASRec | 1.0000 | 0.2233 | 123.91% | 1.0000 | 0.3444 | 53.16% | 1.0000 | 0.4010 | 31.27% |
| **LLM-based** | | | | | | | | | |
| Llama2 | 0.3443 | 0.0246 | 1932.52% | 0.4421 | 0.0421 | 1152.97% | 0.1653 | 0.0135 | 3799.26% |
| ChatRec | 1.0000 | 0.3770 | 32.63% | 0.9895 | 0.2000 | 163.75% | 0.9798 | 0.3626 | 45.17% |
| MoRec | 1.0000 | 0.1652 | 202.66% | 1.0000 | 0.2822 | 86.92% | 1.0000 | 0.3911 | 34.59% |
| TALLRec | 0.9836 | 0.4180 | 19.62% | 0.9263 | 0.3895 | 35.43% | 0.9840 | 0.4637 | 13.52% |
| LLaRA | 1.0000 | 0.4508 | 8.51% | 0.9684 | 0.4421 | 19.32% | 0.9975 | 0.4949 | 6.36% |
| **Ours** | | | | | | | | | |
| iLoRA | 1.0000 | **0.5000** | - | 0.9891 | **0.5275** | - | 0.9981 | **0.5264** | - |

- **RQ3:** What is the impact of the designed components (e.g., the gating network, expert settings) on the recommendation performance of iLoRA?

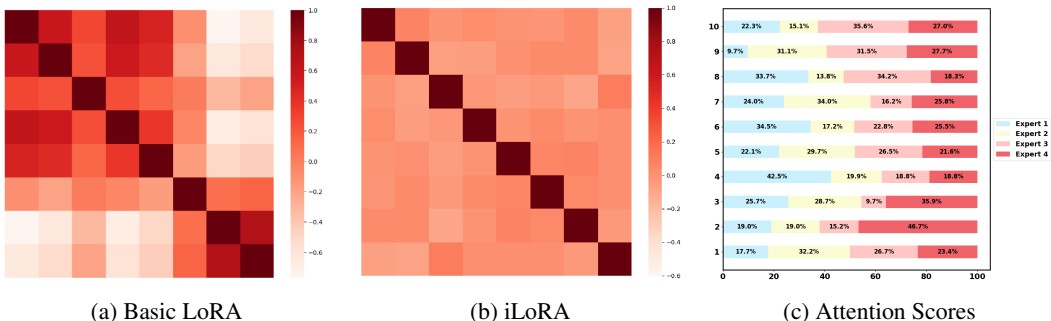

(a) Basic LoRA      (b) iLoRA      (c) Attention Scores

Figure 3: 3a and 3b separately show gradient similarities of LLaRA and iLoRA, with sequences partitioned into 8 clusters; 3c exhibits the attention scores over four experts, for ten sequences.

## 4.1 Investing Rationale of Instance-wise LoRA (RQ1)

We begin by experimenting with LLaRA [9], an LLM-based recommender using a uniform LoRA module, to identify a key limitation: negative transfer between significantly different sequences. Next, we examine the experts within iLoRA, which employs an instance-wise LoRA module, to demonstrate the varying attributions of experts when handling different sequences. For more details, see Appendix B.

### 4.1.1 Negative Transfer in Uniform LoRA & Instance-wise LoRA

Gradient similarity reflects the proximity of recommendation sequences [19]. Here we explore whether using LLaRA to perform recommendations conditioned on different sequences exhibits similar loss geometries and vice versa. To achieve this, we use Euclidean distance to control task similarity and gradient similarity to measure loss geometry. In Figure 1, a symmetric heatmap visually displays the average gradient similarity across all LLaRA checkpoints at different training steps. We demonstrate this test in Figure 3. Specifically, we observe strong clustering along the diagonal of the gradient similarity matrix for sets of recommendation sequences that are closely related in the Euclidean space. Conversely, recommendation sequences that are distant in Euclidean space exhibit correspondingly lower gradient similarity, leading to negative transfer.

In contrast, we visualize iLoRA's gradient similarity among the identical clusters. The similarities between some clusters tend to achieve zero scores, indicating iLoRA's capability to mitigate the negative transfer between significantly different sequences.

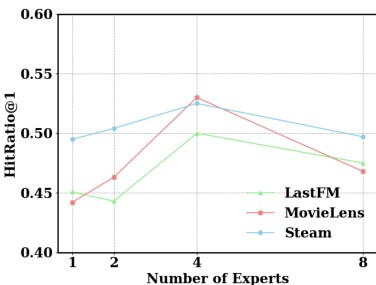
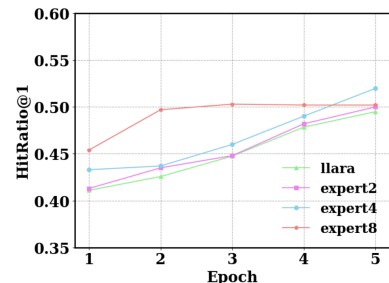

(a) Ablation study on the number of experts.   (b) Comparison of performance over various epochs

Figure 4: 4a illustrates the performance of iLoRA *w.r.t.* HitRatio@1 across different datasets with varying numbers of experts. 4b further demonstrates the HitRatio@1 performance of the model across different epochs during training on the Steam dataset with varying numbers of experts.

### 4.1.2 Expert Showcase in Instance-wise LoRA

In this section, we visualize the attention scores of iLoRA's four experts for ten distinct sequences in Figure 3c. Each horizontal bar represents a sequence, and the length of the segments within each bar indicates the percentage of attention scores assigned to each expert. We have several findings:

- **Sequence Variability:** There is significant variability in expert activation across different sequences. For example, Sequence 4 heavily relies on Expert 1 with a 42.5% activation weight, while Expert 4 only contributes 18.8%, demonstrating distinct preferences for different experts among sequences.

- **Expert Contribution:** Certain experts have notably high contributions for specific sequences. For instance, in Sequence 2, Expert 4 has a dominant activation weight of 46.7%, indicating that this expert captures the personalized preferences of the user group represented by Sequence 2.

- **Collaborative Contribution:** Some sequences exhibit a more balanced distribution of activation weights among multiple experts, suggesting collaborative contributions. For example, in Sequence 9, Experts 2 and 3 have similar activation weights of 31.1% and 31.5%, respectively, indicating their joint influence on the recommendations.

These observations demonstrate that iLoRA effectively adjusts expert activation based on the characteristics of each sequence.

## 4.2 Performance Comparison (RQ2)

This section comprehensively compares iLoRA against some traditional and LLM-based recommenders. We conduct a holistic evaluation, considering metrics of both HitRatio@1 and ValidRatio across LastFM, MovieLens, and Steam datasets to demonstrate the effectiveness of iLoRA. The results of this comparison are summarized in Table 1.

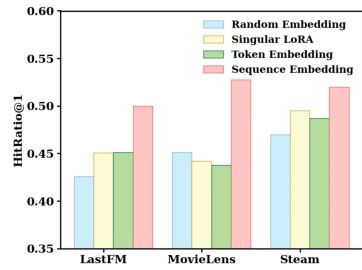

Figure 5: Effects of iLoRA's components

Our findings indicate that iLoRA consistently outperforms these baseline models across the three datasets. Specifically, iLoRA achieves the highest HitRatio@1 metrics of 0.5000, 0.5275, and 0.5264 on the LastFM, MovieLens, and Steam datasets, respectively. These results demonstrate the efficacy of leveraging sequence representations as guidance signals to fine-tune LoRA parameters, enabling personalized recommendations at the parameter level.

## 4.3 Ablation Study (RQ3)

In this section, we analyze the effectiveness of the main components of iLoRA in Section 4.3.1. Subsequently, in Section 4.3.2, we conduct an in-depth investigation and analysis of how varying the number of experts affects the performance of iLoRA.

### 4.3.1 Effects of Gating Network

Here we explore the influence of sequence representation on the gating network and MoE. Going beyond the sequence-tailored representation, we test two variants: using random-initialized and token-collapsed embeddings as the guidance. As Figure 5 shows, using sequence representation as the guidance consistently outperforms the other variants across three datasets, illustrating the rationale of our gating network and the benefits for the MoE combination.

### 4.3.2 Effect of Expert Numbers

In this section, we investigate how iLoRA would react to the number of experts. As depicted in Figure 4a, our model achieves optimal performance when the number of experts is set to 4. Increasing the number of task experts does not necessarily correlate with enhanced performance. Specifically, employing only 2 experts does not significantly improve the HitRatio@1 metrics on Steam and MovieLens datasets, while showing a slight decrease on LastFM. However, with the increase to 4 experts, the model exhibits its best performance across all three datasets, notably surpassing the 2-expert variant. To elaborate, on LastFM, MovieLens, and Steam datasets, the performance of the 4-expert variant exceeds that of the 2-expert variant by 5.2%, 6.4% and 2.2%, respectively. When the number of experts is further increased to 8, the performance resembles that of the 4-expert scenario or even shows a slight decrease. This suggests that the benefits of increased capacity gradually converge as we utilize more experts. Consequently, we adapt 4 experts as the default setting.

Furthermore, we analyzed the performance across different numbers of experts at various epochs, on the Steam dataset. It is evident that as the number of experts is set to 1, 2, and 4, the overall recommendation performance of the model steadily improves with training progress. Under this configuration, the HitRatio@1 values exhibit a positive correlation with the number of experts. However, when the number of experts reaches 8, the data indicate that the model rapidly achieves a decent performance, but subsequent HitRatio@1 values do not show significant improvements with increasing epochs. We speculate that as the number of experts increases to 8, the model may overly focus on personalized user behaviors, leading to a decrease in generalization ability and premature overfitting.

## 5 Conclusion

In this paper, we introduced instance-wise LoRA (iLoRA), a novel fine-tuning framework designed to address the challenges posed by the substantial individual variability in user behaviors within sequential recommendation systems. By integrating the mixture of experts (MoE) concept into the basic LoRA module, iLoRA dynamically adjusts to diverse user behaviors, thereby mitigating the negative transfer issues observed with standard single-module LoRA approaches. iLoRA represents a significant advancement in the application of large language models to sequential recommendation tasks. By incorporating a mixture of expert frameworks within the LoRA module, iLoRA provides a more nuanced and effective means of tailoring recommendations to individual user preferences, paving the way for more personalized and accurate recommendation systems.

## 6 Limitation

While iLoRA demonstrates promising results, there are several limitations to consider. First, our experiments are constrained by computational resources, limiting the exploration of a larger number of expert combinations and their potential impact on recommendation performance. Second, we do not extensively investigate the effects of using hard routing for recommendations with a large number of experts. Finally, our study focused on sequential recommendation tasks, and the applicability of iLoRA to other types of recommendation systems or domains remains to be explored. These limitations suggest that further research is needed to fully understand the scalability and effectiveness of iLoRA with more complex expert configurations.

# 7 Broader Impact

Our proposed method, Instance-wise LoRA (iLoRA), advances sequential recommendation systems by sequence-tailored recommendations. By leveraging the Mixture of Experts (MoE) framework, iLoRA streamlines the user experience, reduces decision fatigue, and promotes inclusivity in online spaces. Its instance-wise adaptation mechanism ensures diverse content exposure, fostering a more enriched online discourse. Beyond recommendations, iLoRA's principles extend to education, healthcare, and e-commerce, offering customized solutions in various domains. Overall, iLoRA represents a step forward in enhancing user experience and promoting inclusivity in the digital landscape.

However, despite the advancements offered by iLoRA, it is essential to acknowledge potential drawbacks. Algorithmic biases present in the training data may persist, potentially amplifying existing biases in recommendations. Moreover, over-reliance on customization could lead to filter bubbles, limiting exposure to diverse viewpoints and serendipitous discovery. Additionally, concerns regarding privacy arise due to the collection and analysis of user data for personalized recommendations, necessitating careful consideration of ethical implications in its deployment.

## Acknowledgments and Disclosure of Funding

This research is supported by the National Natural Science Foundation of China (92270114, 62302321) and the National Science and Technology Major Project (2023ZD0121102). This research is also supported by the advanced computing resources provided by the Supercomputing Center of the USTC.

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

# A Description of Figure 1

This figure illustrates the gradient similarity of LoRA modules across all training steps. We utilize signals from the collaborative space to partition the sequence dataset into 8 clusters. Euclidean distance is employed to evaluate the proximity between clusters, whereas gradient similarity is measured to assess geometric loss. Clusters that are closer in the collaborative space are depicted as closer together in the left side of the figure, with darker-colored cells indicating higher gradient similarity. On the right side of the figure, we conducted a case study on three sets of data with pairwise cosine similarities of 0.86 and -0.75. For sequences containing more than three movies of the same genre, we performed a cross-matching analysis. It was observed that the two user sequences from the cluster with a cosine similarity of 0.86 both exhibited a strong interest in thriller movies, sharing two identical items in their interaction histories at the same time. In contrast, the two user sequences from the cluster with a cosine similarity of -0.75 did not demonstrate any noticeable preference similarities.

# B Experimental Design and Evaluation

**Datasets.** To validate the generalization ability of iLoRA, we conducted extensive experiments on three datasets derived from real-world recommendation scenarios: LastFM, MovieLens, and Steam:

- **LastFM** A popular music dataset for recommendation research, featuring histories of user-artist interactions, user demographic details, artists' names and associated social tags.

- **MovieLens** A widely used benchmark in recommendation systems, containing user ratings and metadata for movies.

- **Steam** A collection of user interaction data from the Steam gaming platform, featuring information on game ownership, playtime, and user reviews.

**Baselines.**

We compared iLoRA with several models, including traditional sequential recommendation models and those based on Large Language Models (LLMs), such as GRU4Rec[28], Caser, SASRec[24], Llama2[21], GPT-4, MoRec, TallRec[5], and LlaRA[9]. GRU4Rec, Caser and SASRec utilize recurrent neural networks, convolutional neural networks, and Transformer encoders, respectively, to capture sequential patterns in user behavior. Llama2, Meta's open-source language model, building on the original Llama to deliver improved performance. We fine-tune the 7B version of the model for our experiments. GPT-4, OpenAI's advanced language model, has held the state-of-the-art position across various tasks for an extended period. We directly utilize its API for our experiments. MoRec uses pre-trained modality encoders to capture item-specific features, improving recommendation accuracy. TALLRec guides LLMs through fine-tuning recommendation corpora. LlaRA further improves the recommendation effectiveness of LLMs by aligning collaborative signals to the text space through a hybrid prompt method.

**Training Protocol.**

In our study, we fine-tune the Llama2-7B [21] model to validate our approach. Experiments for traditional sequential recommendation baseline models are conducted on a single Nvidia A40, while our iLoRA framework is implemented on a single Nvidia A100. All experiments are carried out using Python 3.8 and PytorchLightning 1.8.6.

**Evaluation Protocol.**

To assess the performance of iLoRA and baseline models, we construct candidate sets for each sequence by randomly selecting 20 non-interacted items while ensuring the inclusion of the correct next item. Performance is evaluated using the HitRatio@1 metric, wherein models attempt to identify the correct item from the candidate set. LLM-based recommenders, when provided with appropriate prompts, generate a single candidate item. Conversely, traditional models are adapted to select the item with the highest probability. Additionally, we introduce the ValidRatio metric to quantify the proportion of valid responses (i.e., items within the candidate set) across all sequences. This metric serves to evaluate the models' adherence to instructions accurately.

## C Experimental Setup of RQ1

We extend the previous research setup to train models on multi-task scenarios[51]. Specifically, we jointly train recommendation sequences in a basic LoRA training framework. We use an effective batch sizes of 128 sequences. The recommendation sequences are divided into multiple tasks using representations derived from the sequential recommendation model SASRec, with a dimension of 64.

To investigate large-scale multi-tasking in sequential recommendation tasks, we sample 40k sequences from the Steam dataset. We clustered these sequences into 8 sub-datasets using Euclidean distance. At checkpoints across 1k training steps, we measured the pairwise cosine similarity of model gradients for all sequences. We averaged the LoRA gradients that were bound to the same modules, such as $gateproj$.

## D Related Work

**Large Language Models** Sequential recommendation [24, 32, 31, 52–54] aims to predict the user's next likely item of interest by analyzing their past interaction patterns and aligning with their preferences. Recent years have witnessed a surge of activity in language modeling research, establishing it as a cornerstone for both understanding and generating language. This momentum has given rise to a new breed of language models (LMs), including notable works such as BERT [36], GPT-3 [35], LLama [37], LLama2 [21], Mistral-7B [38], Alpaca [39], and Vicuna [40]. These LMs, predominantly based on the Transformer architecture, have demonstrated remarkable versatility, exemplified by models like BERT [36] and T5 [34], owing to their extensive training corpus. A significant stride in this domain has been the exploration of scaling effects, with researchers pushing the boundaries by augmenting both the parameter and training corpus scales to unprecedented magnitudes, encompassing billions of parameters and trillions of training tokens [37, 21, 40, 35, 55]. These Large Language Models (LLMs) exhibit substantial performance enhancements and showcase unique capabilities, including but not limited to common sense reasoning and instruction following. Moreover, the development of domain-specific LLMs further enriches this landscape. Models tailored to specific domains, such as finance [56], medicine [57], and law [58], amalgamate domain expertise with the inherent commonsense knowledge of general LLMs. These advancements not only broaden the scope of LLM applications but also inspire exploration into their potential utility in recommendation systems.

**Mixture of Experts** Mixture of Experts. MoE models [59, 22, 23] are considered as an effective way to increasing the model capacity in terms of parameter size. In MoE, certain parts of the model are activated while the computation is kept the same or close to its dense counterpart. Recently, it has been thoroughly investigated in the field of computer vision [60, 61], natural language processing [62, 63] and multi-modal learning [64, 65].

## E Statistics

In our experimental settings, we closely follow the setup of prior works [9] to ensure a fair and meaningful comparison. Specifically, for all conventional sequential recommendation baselines, we employ the Adam optimization algorithm, establishing a learning rate of 0.001, an embedding dimension of 64, and a batch size of 256, respectively. Furthermore, we incorporate L2 regularization, and the regularization coefficient is fine-tuned through a grid search over a specified set of possible values.We report the average results from five different random seeds, specifically selected from the set, To reduce the effect of randomness.

For experiments involving methods based on large language models (LLMs), we incorporate a warm-up strategy for the learning rate, which begins the training process with an initial rate set at a fraction of the maximum learning rate. Specifically, the maximum learning rates are set to $2e^{-4}$ for the LastFM dataset and $1e^{-4}$ for the MovieLens and Steam datasets. Notably, the projector parameters are updated using the same learning rate curve integrated with iLoRA during training.

We adopt a cosine annealing scheduler to adjust the learning rate across training steps, which facilitates a smooth decrease in the learning rate, thereby improving stability during training. Additionally, half-precision computation (mixed-precision training) is used to enhance memory efficiency and accelerate the training process.

