# OpenReview forum: "Customizing Language Models with Instance-wise LoRA for Sequential Recommendation"
_NeurIPS.cc/2024/Conference — NeurIPS 2024 poster_

### Official Review · Reviewer_wwLq · 2024-07-05

**Soundness:** 4
**Presentation:** 4
**Contribution:** 3
**Rating:** 7
**Confidence:** 4

**Summary:**

Sequential recommendations is well studied problem with impact across industries which try to personalise a users experience based on their past interactions. Recent popularity of LLMs has led to a lot of research into their use for this task through various generation methodologies. One of those methods is the use of LoRA to adapt the LLM to a specific data distribution (i.e. dataset) leading to better recommendations but it has been shown that due to the high variability in the sequences being input it is often hard to capture all kinds of patterns. This leads to negative transfer of information - to solve which the authors propose a method called iLoRA which is really MoE meets LoRA. Instead of a single LoRA module they propose having multiple LoRA "experts" with the motivation that they will capture different kinds of patterns and information in them. To combine the output from these experts they use an "attention"-like gating mechanism that looks at the input sequence embeddings (created using existing techniques) and computes a softmax to get the weight of each expert to be considered.
The paper contains extensive experimentation to prove out the ideas and contributions and validate the motivation behind the idea. The metrics show that this approach is better than the other SoTA comparable methods in industry.

**Strengths:**

The paper is written clearly, is easy to follow and explains the key ideas that inspired and led to the approach described. The experiments conducted (including ablation studies) provide sufficient evidence to prove the claims and contributions.
The paper is able to take key concepts (MoE, Attention, LoRA, item recommendations) in research and combine them in a new way to solve a problem like sequential recommendation which is significant across many industries.
The results on the metrics being used by the paper outperform other SoTA techniques which proves out the significance for industry use. The performance gain is also demonstrated across 3 popular datasets which gives it authenticity.
The idea is original albeit similar techniques are present in research where people have either tried to a weighted sum of multiple LoRA's or swap them out at runtime based on a gating mechanism. Although I haven't seen an application in recommendation systems yet.

**Weaknesses:**

The authors should have shown the contribution by expanding to more metrics - at the least HitRate@K, but also metrics like NDCG@K.
These are very common metrics in the recommendation space and provide a more holistic view of the output quality.
The novelty in the paper is limited as it combines common research ideas in a not so unique way.
Also, the impact of training data size on the model performance was not studied - recommendation models can be heavily dependent on this due to the generic sparse nature of data so this analysis would give more weight and significance to the paper.
I would have liked to understand more on the out of domain abilities of such a model and also specific abilities on cold start recommendations - which is a big challenge in this space. This was not discussed.

**Questions:**

1. Did you do any comparisons with the idea of training multiple full LoRA's and then doing a weighted sum of them OR swapping them based on instance embeddings ?
2. Other than the overall metrics did you do any analysis on performance when the data is sparse , or situations of cold start recommendations ?

**Limitations:**

The authors have discussed the major limitations. The impact of invalid outputs due to the generative approach was not discussed in detail - this could have potential risks.

---

> ### Author Rebuttal · Authors · 2024-08-07
>
> # Response to Reviewer wwLq
>
> **Comment:**
>
> We gratefully thank you for your valuable comments! Here we meticulously give point-by-point responses to your comments, and further revise our manuscript following your suggestions. We sincerely appreciate the depth of your insights, which have undoubtedly enriched our work. Here we meticulously give point-by-point responses to your comments.
>
> > **Comment 1: More evaluation metrics** - "The authors should have shown the contribution by expanding to more metrics - at the least HitRate@K, but also metrics like NDCG@K. These are very common metrics in the recommendation space and provide a more holistic view of the output quality."
>
> We fully agree that additional evaluation metrics are needed. Inspired by your comment，For the LastFM and MovieLens datasets, we have expanded the evaluation metrics, including HitRatio@3, HitRatio@5, NDCG@3, and NDCG@5, as **Table 1** in one-page uploaded pdf shows.
>
> > **Comment 2: Limited novelty** - "The novelty in the paper is limited as it combines common research ideas in a not so unique way."
>
> Thank you very much for your feedback! In the context of recommendation systems, user preferences are reflected in their historical interaction sequences. Unlike single tokens that lack specific meanings in the recommendation context, user representations encompass rich user preferences and can serve as inputs for gating, outputting instance-wise expert activation weights. while iLoRA draws inspiration from the soft-attention concept, it innovatively guides routing using user interaction history representations, which is a core and critical innovation of our entire iLoRA framework. By routing with user representations, iLoRA effectively provides instance-wise unique activation weights for each different sequence, enabling recommendations that better capture user differences and fine-grained preferences, mitigating the negative transfer.
>
> > **Comment 3: Lack of data size impact experiments**
>
> We appreciate your insightful question regarding the impact of training data size on model performance. Inspired by your comment, we have delved into the influence of the generic sparse nature of data on final recommendation effectiveness. Specifically, on the LastFM dataset, we have conducted experiments at varying degrees of data sparsity, dropping sequences randomly by 25%, 50%, 75%, and 90% at the user level. Subsequently, we have evaluated the performance of both traditional sequence recommendation models (SASRec, Caser, GRU4Rec) and LLM-based recommenders (TALLRec, LLaRA, iLoRA) across these datasets, as illustrated in the **Figure 3** in one-page uploaded pdf.
>
> Several key observations emerged from our study:
>
> - Traditional sequence models exhibit a relatively gradual improvement in recommendation performance as training data size increases. In contrast, LLM-based recommenders demonstrate rapid performance gains with smaller training sets. This phenomenon may stem from LLMs quickly capturing foundational patterns of the recommendation scenario from limited samples, thereby bridging the gap between training data and real-world recommendation scenarios and manifesting "emergent" behavior on unseen data. However, as training data size grows, diminishing marginal returns on model performance become evident, albeit maintaining a positive correlation with data size within our training set.
> - Our proposed iLoRA framework consistently outperforms baseline methods under equivalent training data sizes, underscoring the superior effectiveness of iLoRA in recommendation tasks.
> - Even with only 10% of user training data, all LLM-based recommenders employed in our study outperform traditional sequence recommendation methods in terms of HitRatio@1, highlighting the substantial potential of LLM-based recommendations in advancing traditional recommendation paradigms.
>
> In conclusion, these findings underscore the nuanced relationship between training data size and model performance in recommendation systems, particularly highlighting the unique advantages of LLM-based approaches in sparse data environments.
>
> > **Comment 4: Discussion about out-of-domain and cold-start recommendation abilities**
>
> We value your comments.
>
> To further investigate the out-of-domain capabilities of LLM-based Recommenders, we have conducted an additional experiment, as depicted in the **Figure 4** in one-page uploaded pdf. Specifically, we fine-tuned iLoRA using training data from different domains: 1) iLoRA (LastFM), 2) iLoRA (MovieLens), and 3) iLoRA (Steam). We then evaluated the models on the test sets of LastFM, MovieLens, and Steam. Our results demonstrate iLoRA's ability to generalize across domains. For instance, after fine-tuning exclusively on MovieLens data, 'iLoRA (MovieLens)' exhibited strong performance on the LastFM dataset, even surpassing some traditional sequence models trained and tested directly on LastFM data. This finding is impressive, indicating that iLoRA possesses cross-domain generalization capabilities beyond single-domain adaptation."
>
> > **Comment 5: Comparison with ensemble LoRA models and instance-based swapping** - "Did you do any comparisons with the idea of training multiple full LoRA's and then doing a weighted sum of them OR swapping them based on instance embeddings?"
>
> We appreciate your comments. We considered the approach you suggested, which theoretically could lead to better recommendation performance. However, in practice, it would incur greater training resource expenses. Our iLoRA framework aims to release negative transfer while maintaining a consistent trainable parameter count to mitigate negative transfer. Thus, we have opted to maintain a fixed total parameter count, specifically expert number multiplied by each expert's rank equals 16.

---

### Official Review · Reviewer_7niz · 2024-07-12

**Soundness:** 3
**Presentation:** 3
**Contribution:** 2
**Rating:** 4
**Confidence:** 4

**Summary:**

The author focus on extend sequential recommendation task with the help of large language models. The author proposed instance-wise LoRA and integrate with mixture of experts framework to capture specific interests of user preferences. Experiments results on two benchmark datasets demonstrate the effectiveness of proposed model.

**Strengths:**

1. The authors proposed iLoRA for sequential recommendation task, which address previous works failing to capture individual variability. It creates a diverse array of experts and captures specific aspects of user preferences, guiding the gating network to output customized expert participation weights.
2. Extensive experiments on two benchmark datasets demonstrate the effectiveness of iLoRA. The ablation studies clearly show that every part of the proposed model makes sense.
3. The paper is well-written. Figures and tables are very clear and easy to read.

**Weaknesses:**

See Questions to authors.

**Questions:**

. How do you partition the sequence data into 8 clusters? Clustering by calculating Euclidean distance of what? is the output embeddings of SASRec? Explain why you do that.

2. I think that the proposed mod framework in this manuscript is kind of like the soft-attention mechanism with some parameters having shared weights. What are the in-depth or essential differences between the proposed MoE framework and the soft-attention mechanisms?

3. Lacks further analysis in ablation study section 4.3. The authors just repeat the results that have shown in figures. For example, 4-experts can achieve the optimal performance and 8-experts reduce it. Why? With fewer training epochs, 8-experts outperforms, but it cannot keep it while training epochs increase. Is it because of the hidden dimension r of LoRA that limits the performance of more experts?

4. The writing language can be further checked, since there are minor language mistakes such as nvidia-smi a100.

**Limitations:**

The author explains the limitations of the model constructed in the article, such as the recommendation strategy and universality that still need to be explored.

---

> ### Author Rebuttal · Authors · 2024-08-07
>
> We gratefully thank you for your valuable comments! Here we meticulously give point-by-point responses to your comments, and further revise our manuscript following your suggestions. Hope that our responses can address all your concerns.
>
> ---
>
> > **C1: Questions about experimental details and motivation**
>
> Thank you for your feedback. Existing research assumes that gradient conflicts, defined as negative cosine similarity between gradients, lead to negative transfer [1], where different gradients interfere with each other, affecting model performance [2]. To support our research, we aim to investigate whether training a single model to handle all recommendation sequences could result in negative transfer. Previous research [2] suggests that close tasks enjoy similar loss geometries and vice versa. Therefore, we need to partition the original dataset based on user similarity.
>
> In recommendation scenarios, user preferences are implicitly reflected in their historical interaction sequences. Therefore, we partition the user population using representations derived from their historical interaction records. **SASRec** stands as a cornerstone in sequential recommendation tasks. As you mentioned, we utilize the model to generate rich collaborative information representations for each user. Based on these representations, we apply the **K-means** clustering algorithm to divide the dataset into clusters, with the number of clusters set as a hyperparameter (in this case, 8). Clustering based on **Euclidean distance** and cosine similarity yields similar results, with comparable sequence counts per cluster.
>
> ---
>
> > **C2: Lack of clear distinction from soft attention mechanism**
>
> Thank you for your feedback! As you mentioned, our proposed iLoRA framework draws inspiration from the soft-attention concept, utilizing multiple experts weighted and summed to obtain activation weights. However, the innovation lies in the differences between iLoRA and traditional soft attention methods. In the context of recommendation systems, user preferences are reflected in their historical interaction sequences. Soft attention routing methods typically employ token-level routing or use the entire input *x* as the routing input. A single token lacks specific meanings in the recommendation context, whereas using the entire prompt as routing input introduces considerable redundancy and irrelevant information. Unlike single tokens that lack specific meanings in the recommendation context, user representations encompass rich user preferences and can serve as inputs for gating, outputting instance-wise expert activation weights.
>
> This innovation inspired the development of iLoRA, which innovatively guides routing using representations of user interaction history—a core and critical feature that distinguishes iLoRA from traditional soft-attention mechanisms. By routing with user representations, iLoRA effectively provides instance-wise unique activation weights for each different sequence, enabling recommendations that better capture user differences and fine-grained preferences, thereby mitigating the negative transfer.
>
> To demonstrate the effectiveness of iLoRA compared to traditional soft attention mechanisms, we have conducted an additional ablation experiment. We compared the performance of iLoRA (user representation routing) against two soft attention routing methods (token-level routing and full input *x* routing) on the LastFM dataset, as depicted in **Figure 2** in one-page uploaded pdf. This experiment underscores our innovative use of user historical interaction sequences for routing.
>
> ---
>
> > **C3: Lacks further analysis in ablation study**
>
> We appreciate your insightful observation, which highlights an intriguing phenomenon we have uncovered in our research. Thank you for pointing out the shortcomings in our analysis in section 4.3; we will revise our paper accordingly.
>
> Your observation aligns with a common finding when applying MoE+LoRA in other domains [4-5]. Specifically, increasing the number of experts does not necessarily improve model performance.
>
> We propose the iLoRA framework to enhance model expressiveness while maintaining a constant trainable parameter count, mitigating negative transfer. Therefore, we maintain a fixed total parameter count, i.e., expert number * each expert's rank = 16. Increasing the number of experts benefits by allowing the model to focus more on individual differences between users. Conversely, increasing the hidden dimension *r* of each expert can encourage the model to emphasize universal preferences across the user population, but it may simultaneously limit the expressive power of additional experts. Thus, under the constraint of a fixed total parameter count, this becomes a trade-off scenario. The optimal number of experts likely depends on the data's heterogeneity. In our experimental setup, the model indeed achieved optimal performance with `expert num=4`.
>
> Regarding your second point, we agree with your analysis. A smaller number of experts restricts the iLoRA framework's ability to capture fine-grained user preferences. We appreciate your insightful discussion on this issue.
>
> > **Comment 4: Language mistake**
>
> We sincerely apologize for the oversight and have promptly addressed these issues in the manuscript. We will carefully review the language issues and make corrections in the revised version.
>
> Once again, we appreciate your thorough review and conscientious feedback on our work!
>
> [1] Characterizing and avoiding negative transfer.
>
> [2] Investigating and Improving Multi-task Optimization in Massively Multilingual Models.
>
> [3] LLaRA: Large Language-Recommendation Assistant.
>
> [4] Exploring Training on Heterogeneous Data with Mixture of Low-rank Adapters.
>
> [5] MOELoRA: An MOE-based Parameter Efficient Fine-Tuning Method for Multi-task Medical Applications.

---

> ### Author Response · Authors · 2024-08-13
>
> Dear Reviewer,
>
> Thank you very much for your valuable and constructive feedback. We greatly appreciate the time and effort you have put into providing us with these insights, which have significantly helped us in refining our paper.
>
> Regarding your primary concern about **experimental details and motivation**, we have provided additional discussion and clarification. For the point you raised about the **Lack of clear distinction from the soft attention mechanism**, we have thoroughly explained the differences and advantages of our proposed iLoRA framework compared to the soft attention mechanism. Additionally, we have included an experiment demonstrating the performance benefits of iLoRA over soft attention routing in **Figure 2** in one-page uploaded pdf. Concerning the **Lack of further analysis in the ablation study**, we have added a detailed analysis following your suggestion. We also have throughly examined the linguistic concerns raised and will make the appropriate revisions in the final version.
>
> Thank you once again for your suggestions, and we hope that our response effectively addresses your concerns! We truly appreciate your support, encouragement, and understanding.
>
> Best regards,
> Authors

---

### Official Review · Reviewer_59LV · 2024-07-19

**Soundness:** 4
**Presentation:** 4
**Contribution:** 3
**Rating:** 7
**Confidence:** 3

**Summary:**

The paper introduces iLoRA, which combines LoRA with user representation-guided mixture of expert architecture. The motivation is clear and the writing is good. Extensive experiments are conducted on three public datasets, demonstrating the performance of the proposed method.

**Strengths:**

1. Timely study on large language models and sequential recommendation.
2. The motivation of the proposed method, i.e., should be more personalized LoRA parameters, is clear and convincing.
3. The writing is good and the paper is easy to follow.
4. Experiments are conducted on three public datasets, demonstrating the performance of the proposed iLoRA technique.
5. Code is available during the reviewing phase.

=== Update After the Rebuttal Phase ===

The rebuttal includes new results and clarifications, which are helpful in addressing my concerns. In recent days, I also conducted experiments using the provided code, and the results further verify the paper’s conclusions. As a result, I will raise my rating and vote for acceptance.

**Weaknesses:**

1. The evaluation setting is kind of fragile. For each input sequence, only 20 candidate items are randomly selected as negative samples. In this setting, the results may be influenced a lot by randomness. In addition, if the proposed method only applies to a candidate size of 20 items, the value of this work to the real recommender systems can be doubted.
2. References of MoE are limited. The related works about MoE are quite concise. The author(s) are encouraged to discuss more about how MoE is applied to LLMs and recommendation models.
3. Presentation issues.
    1. Lines 554 - 562 and lines 563 - 570 are duplicated.
    2. Figure 3 (c) is suggested to be replaced by a vector graph to make it more clear.

**Questions:**

Please find the details in "Weaknesses".

---

> ### Author Rebuttal · Authors · 2024-08-07
>
> We sincerely thank you for your time and valuable comments. Your main suggestions about the evaluation setting help us substantiate wide applicability of our proposed iLoRA. To address your concerns, we have detailed our responses point-to-point below.
>
> ---
>
> > **C1: Fragile Setting of Sampled Evaluation**
>
> Thanks so much for your great suggestions! The issue you raised is highly critical, which is a common limitation in current LLM-based recommendation research. Moreover, we are actively exploring methods such as codebook approaches to address full-rank recommendation scenarios. As the following table shows, preliminary experimental results are promising in this regard.
>
> |    | Amazon-Sports |      |    |    |
> | :--- | --- | --- | --- | --- |
> |              | HitRatio@5    | HitRatio@10 | NDCG@5 | NDCG@10 |
> | SASRec       | 0.0324        | 0.0547      | 0.0182 | 0.0247  |
> | Our new work | 0.0596        | 0.0784      | 0.0389 | 0.0477  |
>
> We note your mention of the experimental setting where we select the next item from candidate items. In traditional recommendation research, there are excellent studies that adopt sampling-based evaluation methods.
>
> Inspired by your query about whether our iLoRA training framework is limited to tasks involving 20 candidate items for next-item prediction, we have extended our experiments on the LastFM and MovieLens datasets to evaluate results with 30 and 40 candidate items, respectively. We benchmarked our approach against TALLRec and LlaRA, with detailed results presented in the table below. We conducted five repetitions of our experiments using different random seeds and calculated a p-value, which was less than 0.05, demonstrating the stability of our proposed iLoRA framework.
>
>
> |         | MovieLens                 |         | LastFM                    |                           |
> | :--- | --- | --- | --- | --- |
> |         | HitRatio@1(30 candidates) | HitRatio@1(40 candidates) | HitRatio@1(30 candidates) | HitRatio@1(40 candidates) |
> | TALLRec | 0.3277                    | 0.2833                    | 0.3461                    | 0.2972                    |
> | LlaRA   | 0.3513                    | 0.3250                    | 0.3772                    | 0.3250                    |
> | iLoRA   | 0.4066                    | 0.3666                    | 0.4283                    | 0.3735                    |
>
> Besides, for the LastFM and MovieLens datasets, we have expanded the evaluation metrics, including HitRatio@3, HitRatio@5, NDCG@3, and NDCG@5, as the following table shows.
>
> |         | LastFM     |            |        |        | MovieLens  |            |        |        |
> | :--- | --- | --- | --- | --- | --- | --- | --- | --- |
> |         | HitRatio@3 | HitRatio@5 | NDCG@3 | NDCG@5 | HitRatio@3 | HitRatio@5 | NDCG@3 | NDCG@5 |
> | GRU4Rec | 0.4370     | 0.4964     | 0.3544 | 0.4110 | 0.4831     | 0.5584     | 0.4075 | 0.4702 |
> | Caser   | 0.4445     | 0.4918     | 0.3564 | 0.4232 | 0.4892     | 0.5603     | 0.4134 | 0.4724 |
> | SASRec  | 0.4253     | 0.4792     | 0.3382 | 0.4073 | 0.4256     | 0.5132     | 0.3881 | 0.4239 |
> | TALLRec | 0.6814     | 0.7473     | 0.3900 | 0.4650 | 0.4874     | 0.5408     | 0.4290 | 0.4601 |
> | LlaRA   | 0.7223     | 0.7862     | 0.6016 | 0.6972 | 0.5505     | 0.6189     | 0.4752 | 0.5067 |
> | iLoRA   | 0.7873     | 0.8350     | 0.6894 | 0.7530 | 0.6464     | 0.7084     | 0.5603 | 0.5905 |
>
> ---
>
> > **C2: Limited Reference of MoE**
>
> Thank you for highlighting this valuable point. Following your feedback, we will revise the relevant content in the Related Work section. Due to space constraints, detailed citation information will be included in the References.
>
> Our approach to integrating MoE with LLM considers three perspectives:
>
> - **NLP Perspective**: Mixture-of-Experts modifies feedforward neural network layers into sparse activation experts, significantly expanding model capacity without a substantial increase in computational costs. Recent explorations of MoE have evolved from sample-level to token-level MoE, with most works aiming to scale up model parameters while reducing computational overhead. Our approach differs significantly. We employ a MoE-like structure that nearly preserves total parameter count to address negative transfer in LLM-based recommendation systems.
> - **LoRA+MoE Perspective**: In the era of LLMs, researchers have integrated MoE concepts into PEFT methods to enhance model performance. LoraHub trains multiple LoRA models and selects different LoRA combinations based on data type during inference. MOELoRA improves model efficiency in medical multitasking by incorporating MoE structures. However, these methods require data types as input to the router during training, which necessitates prior knowledge of data types for selecting LoRA combinations during inference. Our approach is distinct as iLoRA is an end-to-end framework that does not require a priori knowledge of data types for inference.
> - **Recommendation Perspective**: Traditional recommender systems, exemplified by MMoE, demonstrate remarkable capabilities in task scenarios. With the advent of large models, methods like OneRec leverage MoE structures to design expert combinations for multi-domain user embedding, integrating collaborative knowledge into LLMs. Our approach innovates by combining MoE principles with PEFT in LLM-based recommendation scenarios, addressing negative transfer effects caused by individual user differences during fine-tuning. We further propose user representation-guided routing for instance-wise recommendations.
>
> ---
>
> > **C3: Typo & Presentation issues**
> Thank you for reviewing our work so conscientiously, and we sincerely apologize for the oversight. We will carefully review the presentation issues and make corrections in the revised version. Additionally, we will review the presentation of the entire manuscript.
>
> Once again, we appreciate your thorough review and conscientious feedback on our work!

---

> > ### Comment · Reviewer_59LV · 2024-08-09
> >
> > Thank you for taking the time to address my concerns. The rebuttal addressed most of them. I’ll raise my rating on "Soundness" and keep my overall rating.

---

> > > ### Author Response · Authors · 2024-08-14
> > >
> > > Dear Reviewer,
> > >
> > > We would like to express our sincere gratitude for your valuable comments. Your insights on the **Fragile Setting of Sampled Evaluation** and **Limited Reference of MoE** have significantly helped us in establishing a more comprehensive and robust evaluation standard for the proposed iLoRA framework, as well as in providing a more thorough discussion of the relevant content.
> > >
> > > We hope that our responses have addressed most of your concerns. If so, we kindly ask if you could consider revising the overall rating of our paper. If there are any remaining concerns, we would be more than happy to further discuss them with you, especially since the rebuttal period is nearing its end. Please let us know if you have any additional questions.
> > >
> > > Thank you once again for your support and understanding!
> > >
> > > Best regards,
> > > Authors

---

### Official Review · Reviewer_cnYP · 2024-07-25

**Soundness:** 3
**Presentation:** 3
**Contribution:** 3
**Rating:** 5
**Confidence:** 3

**Summary:**

The paper addresses the challenge of personalizing language models for sequential recommendation tasks, where user behaviors exhibit significant individual variability. The proposed solution, Instance-wise LoRA (iLoRA), adapts the mixture of experts concept to tailor LLMsfor this variability.

**Strengths:**

The iLoRA framework proposes splitting the standard LoRA module into multiple experts, each capturing specific aspects of user behavior. Key components of this solution include:

- Splitting Low-Rank Matrices into Experts: Dividing the projection matrices into an array of experts to capture different aspects of user preferences.
- Instance-wise Attentions: Using a gating network to generate attention scores for each expert based on the user's historical item sequence. This customization helps in adapting the model more accurately to individual user behaviors.
- Fine-tuning with Hybrid Prompting: Applying the personally-activated LoRA to fine-tune the LLM, thus mitigating negative transfer and enhancing the model's adaptability.

The solution leverages the mixture of experts approach to ensure scalability and efficiency, maintaining the same total number of parameters as the standard LoRA to avoid overfitting.

The starting point of the article's issue is reasonable, and personalized LoRA is an impressive idea.

**Weaknesses:**

#

- The technical contribution of the paper is insufficient. For example, simply splitting the low-rank matrices into multiple experts representing different aspects seems inadequate and has room for expansion. It could be more refined by setting the number of experts based on different user clusters and controlling the low-rank size of the corresponding experts according to the number of users in each category.

- The paper proposes a framework rather than a model. However, in the main experiments, the authors compare it with numerous models instead of fine-tuning-based methods, which is puzzling. Additionally, the framework proposed in the paper seems to be a general one, or are there any challenges that limit it to sequential recommendation? It might be worthwhile to extend beyond sequential recommendation tasks to broader scenarios.

- The effectiveness of this method needs further validation in extremely large-scale datasets or user groups, or with more evaluation metrics.

- It is hoped that the code will be made open-source for careful review. I may adjust my score based on this.

- The reproducibility does not seem to be good enough, and no code link was found

**Questions:**

see above

**Limitations:**

see above

---

> ### Author Rebuttal · Authors · 2024-08-07
>
> We appreciate your comments, which greatly improve our paper. Below we provide the point-to-point responses to address your concerns and clarify the misunderstandings of our proposed method.
>
> ---
>
> > **C1: More Refined by Integrating Experts with Clustering**
>
> Thanks. Based on your suggestions, we have made clarifications and additional experiments:
>
> - **Clarification of Our Technical Contribution**: We agree that, for instance-wise LoRA, a more refined idea will better show its potential. However, our paper focuses on a **simple yet effective implementation** of this idea.
>
> - **Clarification of User Clustering**: The user clustering (in Sec Introduction & Experiments) operates independently from the expert design (in Sec Method).
>   - The role of clustering is to show whether a uniform LoRA causes negative transfer and  iLoRA can mitigate it, where the number of user clusters is a hyperparameter.
>   - The role of experts is to represent different aspects of users, such that their mixture activated with personal attentions makes the LLM specialized in a user instance, where the number of experts is another hyperparameter.
>
> - **Additional Experiment of More Refined Method**: Based on your comments, we conducted additional experiments where the total rank of all experts is 16, while adjusting each expert's low-rank size, as **Fig 1** in one-page uploaded pdf shows.
>
>   - **Stability and Complexity**: Integrating clustering adds complexity and potential instability due to initialization sensitivity and the dynamic nature of user behavior.
>
>   - **Empirical Observations**: Our findings suggest that uniform expert size distribution in our current model approximates the benefits of clustering but with greater simplicity and stability.
>
> ---
>
> > **C2: Missing Comparison with Fine-tuning Methods & Generalization in Broader Scenarios**
>
> Thanks. Below we clarify the misunderstanding of compared baselines and show the results of additional experiments.
> There are various parameter-efficient fine-tuning strategies [1], such as LoRA, adapters, and prefix tuning.
>
> - **Fine-tuning Baselines**: We compared iLoRA with the fine-tuning baselines (TALLRec, LLaRA). Specifically, TALLRec tunes LoRA on text-only prompts, and LLaRA tunes LoRA along with a simplified adapter on hybrid prompts.
> - **Additional Experiments of More Fine-tuning Baselines**: We added an additional baseline, which adopts prefix-tuning for recommendation.
>
> We visualized the fine-tuning paradigms in **Fig 5** in the one-page uploaded PDF and present their results in the table below:
>
> |           |                    | LastFM     |             | MovieLens  |            | Steam      |            |
> | :-------- | ------------------ | ---------- | ----------- | ---------- | ---------- | ---------- | ---------- |
> |           |                    | ValidRatio | HitRatio @1 | ValidRatio | HitRatio@1 | ValieRatio | HitRatio@1 |
> | TALLRec   | LoRA               | 0.9836     | 0.4180      | 0.9263     | 0.3895     | 0.9840     | 0.463      |
> | PrefixRec | Prefix Tuning      | 0.9627     | 0.3754      | 0.8836     | 0.2973     | 0.9273     | 0.4108     |
> | LlaRA     | LoRA + Adapter     | 1.0000     | 0.4508      | 0.9684     | 0.4421     | 0.9975     | 0.4949     |
> | iLoRA     | instance-wise LoRA | 1.0000     | 0.5000      | 0.9891     | 0.5275     | 0.9981     | 0.5264     |
>
> Clearly, over the three datasets, iLoRA outperforms the baselines using various fine-tuning strategies (LoRA, LoRA+adapter, prefix-tuning) in terms of the HitRatio@1 metric.
>
> - **Generalization of iLoRA to Broader Scenarios**: We are extending the testing of iLoRA on the collaborative filtering task. Due to the time constraints of the rebuttal phase, full results are still pending. We plan to present comprehensive findings during the discussion phase.
>
> ---
>
> > **C3: Evaluation on Larger Datasets & More Metrics**
>
> Thanks. We clarify that the Steam dataset tested in our paper is one of the largest datasets currently used in the LLM-based recommendation domain. Following your suggestions, we are running additional experiments on larger datasets. However, due to the time limit of the rebuttal phase, we haven't collected all the results and will update more results during the discussion phase if you are interested.
>
> We agree that additional evaluation metrics are needed. Beyond HitRatio@1, we added HitRatio@K and NDCG@K as the metrics, where K=3 and 5. The table below demonstrates that LoRA (iLoRA) outperforms other baselines across these metrics, in the LastFM and MovieLens datasets.
>
> |         | LastFM     |            |        |        | MovieLens  |            |        |        |
> | :------ | ---------- | ---------- | ------ | ------ | ---------- | ---------- | ------ | ------ |
> |         | HitRatio@3 | HitRatio@5 | NDCG@3 | NDCG@5 | HitRatio@3 | HitRatio@5 | NDCG@3 | NDCG@5 |
> | GRU4Rec | 0.4370     | 0.4964     | 0.3544 | 0.4110 | 0.4831     | 0.5584     | 0.4075 | 0.4702 |
> | Caser   | 0.4445     | 0.4918     | 0.3564 | 0.4232 | 0.4892     | 0.5603     | 0.4134 | 0.4724 |
> | SASRec  | 0.4253     | 0.4792     | 0.3382 | 0.4073 | 0.4256     | 0.5132     | 0.3881 | 0.4239 |
> | TALLRec | 0.6814     | 0.7473     | 0.3900 | 0.4650 | 0.4874     | 0.5408     | 0.4290 | 0.4601 |
> | LlaRA   | 0.7223     | 0.7862     | 0.6016 | 0.6972 | 0.5505     | 0.6189     | 0.4752 | 0.5067 |
> | iLoRA   | 0.7873     | 0.8350     | 0.6894 | 0.7530 | 0.6464     | 0.7084     | 0.5603 | 0.5905 |
>
> ---
>
> > **C4 & 5:  No Open-Source Code Link Available**
>
> Thanks. Please find our codes in the supplementary material. Following your suggestions, we have uploaded the codes, datasets, and checkpoints to an open-source repository accessible via the anonymous link: https://anonymous.4open.science/r/iLoRA-8C57.
>
> [1] Towards a Unified View of Parameter-Efficient Transfer Learning. ICLR 2022.

---

> > ### Comment · Reviewer_cnYP · 2024-08-13
> > **To Authors**
> >
> > Thank you for your detailed rebuttal, which solved most of my concerns. I am willing to raise the rating to boarderline accept.

---

> ### Author Response · Authors · 2024-08-13
>
> Dear Reviewer,
>
> Thank you very much for your valuable and positive feedback. We appreciate your recognition of our efforts in addressing your concerns, and we are encouraged by your comments that our responses have effectively addressed the issues raised. This motivates us to continue advancing the field with our research.
>
> Following your suggestion, we will incorporate the additional insights discussed in the rebuttal into the revised version of the paper. Thank you once again for your supportive and understanding comments.
>
> Best regards,
> Authors!

---

### Author Rebuttal · Authors · 2024-08-07

# Summary of strengths acknowledged by the reviewers and the responses to address their concerns

**Comment:**

Dear ACs/SACs/PCs,

We would like to summarize the strengths of this work acknowledged by the reviewers, and the responses we have made to address all the reviewers’ concerns.

------

**Strengths** acknowledged by the reviewers:

1. Novelty: timely study (Reviewer **59LV**); address previous works failing to capture individual variability (Reviewer **7niz**); take key concepts in research and combine them in a new way to solve a problem (Reviewer **wwLq**);  mitigating negative transfer and enhancing the model's adaptability(Reviewer **cnYP**)
2. Well-written (Reviewer **59LV**,**7niz**, **wwLq**); Clear and convincing motivation (Reviewer **59LV**);
3. Extensive experiments (Reviewer **59LV**, **7niz**, **wwLq**).

------

There are four main concerns raised by reviewers:

1. More evaluation metrics (Reviewer **cnYP**, **59LV**, **wwLq**).
2. More explanations related to experiments (Reviewer **cnYP**, **7niz**).
3. Inclusion of the impact of training data size (Reviewer **cnYP**, **wwLq**).
4. Inclusion of out of domain abilities and specific abilities on cold start recommendations (Reviewer **wwLq**).

All of these main concerns have been successfully addressed during the rebuttal phase, and we hope that the improvements we made during this stage will be taken into consideration.

------

We sincerely appreciate your valuable time!

Thanks and regards,

Authors

---

### Decision · Program_Chairs · 2024-09-25

**Decision:**

Accept (poster)

**Comment:**

All reviews somewhat borderline, though with 3/4 leaning somewhat positive, mostly willing to recommend acceptance. Reviewers praise the motivation and framework, writing, "timeliness" of the work, and the experiments. More critical comments center around the overall novelty of the work (raised by several reviewers), and the "fragility" of the experiments. These are arguably fairly serious considerations, which make the paper *somewhat* more borderline, but still leaning positive on the basis of the scores, and fairly detailed rebuttal.